# Investigating the Role of Zinc in Atherosclerosis: A Review

**DOI:** 10.3390/biom12101358

**Published:** 2022-09-23

**Authors:** Tong Shen, Qing Zhao, Yumin Luo, Tao Wang

**Affiliations:** 1Department of Neurology, Institute of Cerebrovascular Diseases Research, Xuanwu Hospital of Capital Medical University, No. 45 Changchun Street, Beijing 100053, China; 2Peking Union Medical College, No. 9 Dongdansantiao Street, Beijing 100730, China; 3Beijing Institute for Brain Disorders, No. 10 Xitoutiao, You An Men, Beijing 100069, China; 4Department of Neurosurgery, China International Neuroscience Institute, Xuanwu Hospital of Capital Medical University, National Center for Neurological Disorders, No. 45 Changchun Street, Beijing 100053, China

**Keywords:** zinc, atherosclerosis, zinc homeostasis-associated proteins, endothelial cells, vascular smooth muscle cell, immune cell, cardiovascular risk factors, animal study, human study

## Abstract

Zinc, an indispensable micronutrient for human health, might play an important role in the development of atherosclerosis. Zinc could be involved in the atherogenic process through interaction with atherogenic cells, such as endothelial cells (ECs), vascular smooth muscle cells (VSMCs), and immune cells. In addition, zinc also exerts important positive or negative functions in various atherosclerosis-related risk factors, including lipid metabolism, glucose metabolism, and blood pressure. Currently, evidence focusing on the relationship between zinc status and atherogenic risk factors has been well established, while the direct interaction between zinc and atherosclerosis has not been fully understood. In this review, we aimed to summarize the association between zinc and atherosclerosis and explore current findings on how zinc and zinc homeostasis-associated proteins act in the atherogenic processes.

## 1. Introduction

Zinc, an abundant micronutrient, plays a vital role in numerous biochemical pathways in cells and is indispensable to all creatures. Following iron, zinc is the second most abundant transition metal ion in all living organisms [1]. The human adult body contains approximately 1.4–2.3 g of zinc [2]. Zinc is also a broadly distributed metal ion in the human body, with approximately 85% of zinc concentration found in muscle and bones, 11% in skin and liver, and the remaining 4% in other tissues [3]. The distribution and homeostasis of zinc in the body are highly regulated by various zinc homeostasis-associated proteins, including Zn transporters (ZnT), Zrt- and Irt-like proteins (ZIP), and metallothioneins (MTs). Zinc conducts multiple crucial biological functions in organisms. Firstly, zinc is a redox-neutral metal ion and can stably bind to practically 10% of the total human proteome and still maintain its structural integrity [4]. Secondly, zinc is required in the catalysis and co-catalysis of more than 300 enzymes, which control various metabolic cellular processes [5]. Thirdly, zinc, being “the calcium of the 21st century”, functions as a signaling mediator and targets multiple enzymes involved in cellular signaling, such as kinases and phosphatases [6,7].

Considering the critical biological roles that zinc plays in human health, even a minor zinc deficiency could be disastrous. Multiple abnormalities can cause zinc deficiency, including inadequate intake, decreased absorption, and increased loss. Zinc deficiency is estimated to affect nearly two billion people worldwide, with the estimated prevalence of suboptimal zinc status ranging from 4% to 73% in different countries [8]. Severe zinc deficiency is associated with profound consequences, such as growth retardation and death [7]. Studies estimate that zinc deficiency may be associated with a 4% morbidity and mortality rate of children in developing countries. Even marginal zinc deficiency can cause a variety of pathological conditions, especially age-related pathologies, such as atherosclerosis [7]. 

Atherosclerosis is a pathological condition characterized by atherosclerotic plaque formation in the vascular intima of large and medium-sized arteries. Given the popularity of the western lifestyle, the worldwide prevalence of atherosclerosis is gradually increasing and might develop into an epidemic in the future [9]. Atherosclerosis is the principal cause of cardiovascular diseases (CVDs), including coronary artery disease (CAD), stroke, and peripheral arterial disease. Among adults aged ≥ 20 years in the US, 37.4% of men and 35.9% of women have some form of CVDs, and the estimated prevalence of CVDs in the US adult population could reach 43.9% by 2030 [10]. The identification of risk factors for atherosclerosis and CVDs is crucial for its prevention and treatment. Recent evidence suggests that zinc deficiency plays a vital role in the development of atherosclerosis [11]. 

In this review, we aim to summarize the current findings on the association between zinc and atherosclerosis and investigate the potential effect of zinc and zinc homeostasis-associated proteins in atherogenic processes. Since zinc homeostasis-associated proteins are critical for the physiology and homeostasis of zinc, we first provide a summary of zinc homeostasis-associated proteins and their functions.

## 2. Zinc-Related Proteins

As previously mentioned, zinc plays a crucial role as a structural, catalytic, and regulatory component of many metabolic processes; therefore, it is necessary to clarify the mechanism of zinc homeostasis. In the human body, serum zinc takes up only 0.1% of total zinc, the majority of which is bound to proteins, such as albumin and α2-macroglobulin, with only a small fraction existing as free ions [12,13]. The distribution of cellular zinc is the cytoplasm (50%), nucleus (30–40%), and cell membrane (10%) [13]. Cellular zinc is stored in compartmentalized regions, such as organelles and vesicles, or bound to metalloproteins/metalloenzymes and MTs. Consequently, the concentration of cytosolic free zinc is likewise considerably low, ranging between picomolar and nanomolar [14,15]. Through the release or combination of zinc within proteins and organelles, organisms can cope with transient changes in zinc fluctuation and maintain zinc concentration at a stable level. The four pools of cellular zinc [16] are as follows: (1) zinc that is tightly conjugated to metalloproteins/metalloenzymes as a structural component or cofactor; (2) zinc that binds to MTs with low affinity, which executes the functions as both receptor and donor of zinc in case of zinc perturbation; (3) zinc that is stored in organelles and vesicles and serves as zinc provider through zinc transporters; (4) cytosolic-free zinc that participates in signaling transduction [17]. The last three pools act as labile zinc pools and play a key role when zinc status fluctuates.

The underlying mechanisms of cellular zinc homeostasis maintenance are known as “zinc buffering” and “zinc muffling” [18]. Buffering is realized by cytosolic zinc-binding proteins to obtain consistent levels of free zinc ions in a way that is similar to proton buffering [19,20]. In parallel, the muffling mechanism is instrumental in the instance of transient changes with higher zinc loads in the cytosol and functions by mobilizing the zinc-binding proteins and zinc transporters to move zinc ions into sequestrated subcellular compartments or out of cells [18]. When cellular zinc concentration fluctuates, metal response element-binding transcription factor-1 (MTF-1) can regulate the transcription of MTs and several ZnT transporters [21,22]. The mutual interaction between zinc concentration and transcription factors contributes to the maintenance of cellular zinc homeostasis (see Figure 1). MTs and the zinc transporter family play a pivotal role in not only controlling cellular zinc homeostasis but also the progression of cardiovascular diseases.

### 2.1. ZnT/ZIP

Since the concentration of labile zinc ion is exceptionally low, subtle changes in labile zinc can cause tremendous alternation in cell function. The distribution of zinc in eukaryotic cells is characterized by compartmentalization and is rigorously controlled by membrane zinc transporters, which restricts the availability of zinc ions and provides better buffering ability toward zinc loads [19]. Distinct from iron, more than 30 proteins are involved in the storage and transportation of zinc rather than ferritin alone, which attributes to the deficiency in humoral mediators of zinc status [13]. In humans, 10 members of the ZnT/solute carrier 30A (SLC30A) family were recognized to function in zinc efflux from the cytosol to extracellular space or intracellular organelles, whereas 14 members of the ZIP/solute carrier 39A (SLC39A) family function in mobilizing zinc in a converse way [23]. ZnTs and ZIPs are responsible for the development of atherosclerosis.

ZnT transporters are zinc-proton antiporters residing in the inner membrane of subcellular compartments, except for ZnT-1, which is located in the plasma membrane [24]. When zinc ion conjugates with the metal-binding sites of ZnT protein, it triggers structural changes in the protein and causes conformational transformation, extruding zinc out of the cytosol [13]. ZnT-1 and ZnT-4 have a potential association with CVDs due to their participation in vascular physiology [25]. ZIP transporters, in contrast, are mainly localized to the plasma membrane [26,27] and possibly work as a co-transporter with protons as well [28]. Twenty-four ZIP transporters and twelve MTs are recognized within human heart muscle tissues [29,30], indicating the connection between ZIP proteins and CVDs. A sophisticated interaction exists between ZnTs, ZIPs, and MTs by regulating gene expression reciprocally in response to zinc homeostasis [31]. The function of zinc transporters may be regulated by protein phosphorylation and may thus be linked to many diseases [32].

### 2.2. MTs

MTs are low molecular metal-binding proteins that have selective binding capacity to zinc, cadmium, and copper ions [33]. Human MTs have 11 functional isoforms, which are divided into four subgroups, MT-1 to MT-4 [16]. MT-1 and MT-2 are ubiquitously expressed in many organs and tissues and are engaged in zinc homeostasis, while MT-3 and MT-4 are cell-specific. Each molecule of MTs has up to seven ion-binding sites, attributing to its distinct affinity to zinc and other divalent metals [34]. 

MTs function is crucial for stabilizing the fluctuation of intracellular zinc concentration. Thioneins are translated in the ribosome and chelate with cytosolicfree zinc ion, synthesizing metallothioneins to lower zinc concentration. Conversely, when cellular zinc level is insufficient to maintain MT protein, it rapidly proteolyzes and releases conjugated zinc to keep intracellular zinc concentration stable [35,36]. MT is a potent electrophilic scavenger and cytoprotective agent [37,38] and plays a critical role in immune protection and signal transduction [39,40]. Experiments show that cytokines can either up or down-regulate the expression of MTs, and MTs, in turn, affect the production of cytokines [41,42,43]. In addition, MTs overexpression exerts a positive effect on alleviating diabetes and diabetic cardiovascular complications through attenuating oxidative stress [11,44,45], whereas another study indicated a negative correlation between MT-1 overexpression and insulin secretion [46].

### 2.3. Zinc Finger Proteins

Zinc finger proteins (ZFPs) are the most abundant proteins in human genome [47], which, as the name suggests, are characterized by their zinc finger motifs. Many ZFPs function as transcription factors by binding to specific DNA sequences. The most famous ZFPs in atherosclerosis might be the Krüppel-like factor (KLF) family. KLF2 and KLF4 are assumed to play a central role in maintaining endothelial cell homeostasis and vascular health. Multiple studies have confirmed the atheroprotective effect of KLF2 and KLF4 through inhibition of inflammatory response [48,49,50,51,52]. The activator of KLF2 presented anti-inflammatory effects and attenuated the adhesion of monocytes to endothelial cells, which could be a promising therapeutic strategy for atherosclerosis [53]. Other ZFPs, such as KLF14, zinc finger E-box binding homeobox 1 (ZEB1), and early growth response gene-1 (Egr-1), also participate in the regulation of atherogenesis [54,55,56].

## 3. Zinc and Atherogenic Cells

Zinc plays a direct role in atherosclerosis through the interaction with atherogenic cells, such as endothelial cells (ECs), vascular smooth muscle cells (VSMCs), and immune cells. This process is primarily realized by the regulation of cellular functional molecules. Here, we elaborate on the direct interaction between zinc and proatherogenic cells.

### 3.1. Zinc and Vascular Cells

Atherosclerosis is the most common cause of CVDs, which are characterized by lesions to the arterial wall, subsequent subendothelial plaque formation, and vessel occlusion. It is an artery-centered disease that eventually results in multiple organ damage. Vascular cells, including ECs and VSMCs, are major contributors to the initiation, development, and progression of atherogenic process.

#### 3.1.1. Zinc and ECs

ECs are the innermost vascular layer and function as a barrier between the bloodstream and the vascular walls. Endothelial cell dysfunction (ECD) is a crucial link in the atherogenic process. Early speculation on the role of ECD in atherosclerosis was referred to as the “Response-to-Injury Hypothesis”, which focused on the loss of integrity in the intima of the arterial wall. However, careful examination of the fatty streak lesions in diet-induced animal models indicated no sign of intimal injury or platelet adhesion [57]. With the expansion of understanding of ECD, accumulating evidence suggests that the changes of hemodynamic forces in athero-susceptible regions induce differential gene expression, alternation in cell types, and focal ECD [58]. Unidirectional, high laminar shear stress can upregulate the expression of anti-inflammatory, antioxidative, and antithrombotic genes, such as KLF2, KLF4, and nuclear factor (erythroid-derived 2)–like 2 (NRF2), while in contrast, low shear stress exerts proatherogenic functions by initiating multiple vascular events, including focal permeation, ROS generation, NF-κB activity, and expression of receptors of adhesion molecules, and cytokines that recruit leukocytes [57,59,60]. A former study indicated the upregulation of MTs and ZnT under low shear stress and an increase in free zinc under high shear stress in human umbilical vein endothelial cells (HUVECs) [61]. The dysfunction of ECs results in focal permeation under an intact endothelium and the trapping of circulating lipoproteins in the subendothelial space, thereby initiating the complex atherogenic process [62]. ECD also plays a role in the complications of atherosclerosis, such as vascular occlusion and plaque rapture, through the chronic lesion towards the endothelium [63]. Recent studies suggested that zinc is essential for the proper functioning of ECs through different pathways, as mentioned below [64]. 

Firstly, zinc regulates the synthesis of nitric oxide (NO) in ECs. NO, an endothelium-derived vasodilator, is synthesized via endothelial NO synthase (eNOS). The eNOS-derived NO critically influences the function of ECs and exerts multiple anti-atherosclerotic effects, such as vasodilation, suppression of VSMCs proliferation, and inhibition of leucocyte adhesion, platelet adhesion, and platelet aggregation [65]. The decreased availability of NO, resulting from the reduced expression or activity of eNOS, is actively involved in the atherogenic process [66]. The eNOS is catalytically inactive in its monomeric form and only exerts catalytic function as dimeric eNOS. The dimerization of eNOS is mediated by zinc, located at the dimer interface [25]. N, N, N, N-Tetrakis (2-pyridylmethyl)-ethylenediamine (TPEN), a zinc chelator, was shown to convert eNOS dimers into monomers in ECs [67]. This finding indirectly suggests that zinc is critical for the dimerization of eNOS. Studies found that dietary zinc deficiency during the fetal life, lactation, and postnatal life of rats reduced the expression and activity of Enos [68]. Zhuang et al. [69] further found that zinc supplementation effectively enhances intracellular NO production by elevating the expression and enzymatic activity of eNOS. NO participates in regulating the homeostasis of intracellular zinc in ECs. MTs, proteins functioning as buffer for intracellular-free zinc, release zinc when stimulated by NO, thereby increasing labile zinc level in ECs [25]. This may create a positive feedback loop since the released zinc could then promote the dimerization and activation of eNOS and amplify the anti-atherosclerotic effect of NO. Except for being synthesized by eNOS, NO in ECs can also be produced by inducible NOS (iNOS). Unlike eNOS, iNOS is only expressed under pro-inflammatory conditions. Evidence suggests that iNOS is detrimental in diseases because of its capability of producing excessive NO in response to inflammatory signals [70,71]. Miyoshi et al. [72] indicated that genetic deficiency of iNOS in ApoE-deficient mice reduced the formation of atherosclerotic lesions. Cortese-Krott et al. [73] have shown that zinc down-regulates the expression and activity of iNOS in ECs, and, therefore, limits the undesirable influence of the high levels of iNOS-derived NO.

Secondly, atherosclerosis is considered a chronic inflammatory disorder, while zinc exhibits anti-inflammatory effects in ECs. Faced with varieties of stimuli, ECs express multiple cytokines, chemokines, and cell adhesion molecules, recruiting leukocytes and facilitating their migration to the neointimal space of the vascular wall. Activated ECs could trigger chemokine secretion and recruit the circulating monocytes to the vascular wall, known as monocyte-EC interaction, which is a key event in the formation of atherosclerotic lesions [74]. ZEB is a key member of the zinc finger-homeodomain family. Studies indicated that the overexpression of ZEB1 could decrease the expression of C-X-C motif chemokine ligand 1 (CXCL1), which was a chemoattractant that promoted the monocyte–ECs interaction [75]. These results suggested that ZEB1 inhibits the monocyte–ECs interaction, thereby preventing the aggravation of atherosclerosis. Vascular cell adhesion molecule-1 (VCAM-1), an important cell adhesion molecule, triggers monocytes’ attachment to ECs and promotes the development of atherosclerotic plaques. Zinc deficiency increases the production of VCAM-1 and intercellular adhesion molecules (ICAM-1), thereby promoting inflammation [76]. Zinc additionally affects the expression of multiple pro-inflammatory cytokines in ECs. Zinc deficiency could exacerbate chronic inflammation in ECs by permitting the expression of inflammatory genes [73]. A recent systematic review indicated that zinc status was inversely associated with the development of endothelial inflammation or atherosclerosis [76]. Multiple signaling pathways of various inflammatory cytokines converge with the release of nuclear transcription factor kappa B (NF-κB) [77]. Atherosclerosis is a chronic inflammatory disease of the arterial wall, and the activation of NF-κB has been shown to be involved in all stages of the development of atherosclerosis through mediating downstream canonical and non-canonical signaling pathways [78,79]. Previous studies indicated that zinc deficiency upregulates the activity of NF-κB [80], while zinc supplementation suppresses its activity in ECs [69]. KLF family is another important category of zinc finger transcription factors. Studies indicated that KLF2, KLF4, and KLF11 all conferred important anti-inflammatory actions in ECs by inhibiting the activation of NF-κB [81]. The transactivation activity of peroxisome proliferator-activated receptor (PPAR), the inhibitor of NF-κB signaling, also requires adequate zinc. Zinc deficiency down-regulates the expression of PPARα and PPARγ in cultured ECs [82,83], which then upregulates the DNA binding activity of NF-κB. A20, a protein act as a general inhibitor of NF-κB activation. A positive association between zinc levels and A20 protein concentration was reported in human ECs [84]. This finding suggests that zinc could also suppress the generation of NF-κB-regulated inflammatory cytokines by inducing A20. 

Thirdly, oxidative stress contributes to the development and progression of atherosclerosis, while zinc exhibits anti-oxidative actions in ECs. Oxidative stress is the result of the imbalance between reactive oxygen species (ROS) generation and the antioxidant system of the body [85]. In cultured ECs, zinc deficiency increases oxidative stress, while subsequent zinc supplementation blocks this effect [80,82]. Further studies indicated that zinc deficiency could significantly aggravate oxidative stress by increasing the level of ROS and decreasing antioxidants in the lung and kidneys of mice [86,87]. A recent meta-analysis indicated that zinc supplementation significantly decreases the total antioxidant capacity, as well as malondialdehyde, which is an important metabolite of lipid peroxide [88,89]. Zinc is not redox-active under physiological conditions. Thus, the anti-oxidative action of zinc is realized through indirect mechanisms, such as participation in the synthesis of antioxidative enzymes or working as their catalyzers [90]. Zago et al. [91] found that the effect of zinc in protecting the cell membrane from lipid oxidation may be realized by competition with other redox-active metals for negative charges in the lipid bilayer.

Finally, ECs apoptosis impaired the barrier function of the endothelium and enables the infiltration of circulating leukocytes and lipoproteins in the vascular wall, initiating the atherogenic process [92]. Tricot et al. [93] noted that the turnover and apoptosis of ECs increased in the endothelium of human atherosclerotic plaques. These observations highlighted the important role of the apoptotic cell death of ECs in the development of atherosclerosis. Therefore, suppressing the apoptosis of ECs may be an attractive approach for realizing anti-atherosclerotic effects. ZEB1 could inhibit the apoptosis and senescence of ECs via reducing p21 protein, protecting ECs from oxidative stress-induced dysfunction [94]. Thambiayya et al. [95] found that the decrease in zinc level may contribute to the lipopolysaccharide (LPS)-induced apoptosis. These authors further discovered that the increase in intracellular labile zinc conducts essential functions in the resistance to LPS-induced apoptosis mediated by ZIP-14 or NO [96]. Caspases are pivotal enzymes in the apoptotic pathways, and studies indicate that zinc inhibits caspases [97]. Compared with control cultures, ECs with zinc-deficient cultures were associated with the upregulation of caspase-3 and increased cytokine- or lipid-induced apoptotic cell death [98]. These authors further clarified that subsequent zinc supplementation significantly blocks the activation of caspase-3. Likewise, Fanzo et al. [99] found that the activity of caspase-3 in human aortic ECs was inversely associated with the zinc level in the culture media. Zinc can also inhibit other apoptotic regulators beyond caspase-3, such as caspase-6, caspase-9, and calcium-magnesium-dependent endonuclease [100,101]. 

#### 3.1.2. Zinc and VSMCs

VSMC is a type of smooth muscle cell that is in the media layer of elastic arteries and is responsible for arterial contraction and the production of extracellular matrix (ECM) proteins. VSMCs are the main contributor to atherosclerotic plaque composition. Studies suggest that 30–70% of the plaque cells express VSMC-specific molecules in murine models of atherosclerosis [102,103,104]. The historical view of VSMCs in atherosclerosis typically assumes that VSMCs play an entirely beneficial role in advanced plaques by generating ECM to form the fibrous cap, which stabilizes the vulnerable plaques, preventing them from rupturing. However, genetic lineage tracing studies reveal a more comprehensive and integral perspective on VSMCs function in atherogenesis [102,103,104,105]. During the progression of atherosclerosis, VSMCs undergo de-dedifferentiation and “phenotypic switching” [104,106], which, specifically, refers to the transformation from a contractile phenotype to a proliferative phenotype, which is considered an essential part of the development of atherosclerosis, as well as the development of endothelial damage and lipid deposition [107]. VSMCs in the plaque exhibit a co-expression of both VSMC- and macrophage-specific biomarkers, and the concomitant macrophage properties, such as secreting pro-inflammatory molecules and inducing the migration of inflammatory cells [104,106]. These findings indicate a potential dual role of VSMCs in atherogenesis, and zinc might influence the progress by regulating the function of transcription factors containing zinc-finger structures, such as KLF4 and ZEB1 [55,81,108].

Zinc is associated with VSMC apoptosis and proliferation according to studies conducted by Keith Allen-Redpath et al. [109] and Ethel H. Alcantara et al. [110]. VSMCs from marginally zinc deficient diet rats present with an increased apoptotic effect in the carotid artery and aorta, which is not observed in the endothelial cell layer. The underlying mechanism of interchange to the apoptotic phenotype could be related to increased expression and dephosphorylation of Bcl-2-associated death promoter protein (BAD) (a member of the pro-apoptotic Bcl-2 family), and the decreased activation of the pro-survival extracellular signal-regulated kinase 1/2 (ERK1/2) pathway, which maintains the phosphorylation of BAD. An in vitro study indicates that the apoptotic effect induced by zinc deficiency is potentially mediated by a type of humoral factor since the VSMCs incubated by plasma from zinc-deficient rats also show increased apoptosis [109]. Ou et al. [111] confirmed this hypothesis and demonstrated that zinc not only regulated gene expression directly but also affected the bioactivity of a low-molecular-weight humoral factor. This factor is potentially associated with immune functionality due to the responsive actions of cytokine pathways, but the derivation of this factor has not yet been confirmed. Another study indicates that chronic zinc deprivation can accelerate the proliferation of VSMCs via decreased mitogen-activated protein kinase/c-Jun N-terminal kinase (MAPK/JNK) signaling pathway (apoptotic) activation [110]. Zinc may also play a role in preventing VSMC calcification [112]. Kuzan et al. [113] used the Inductively Coupled Plasma Optical Emission Spectrometer (ICP-OES) to determine the content of microelements in the human aorta, however, the correlation between serum zinc and the severity of atherosclerosis was not observed. Zinc could play a role in vasodilation, via cofunction with calcium. In conclusion, further research is required to establish a correlation between zinc, VSMCs, and atherogenesis.

### 3.2. Zinc and Immune Cells

Atherosclerosis is known as a chronic inflammatory condition of the arterial walls initiated by the deposition of subendothelial lipids. The immune reaction is recognized as a pivotal part of the atherosclerotic mechanism [114]. During the atherogenic process, oxidized low-density lipoprotein (oxLDL) accumulates in the intima and is engulfed by macrophages which later transform into an inflammatory phenotype and turn into foam cells [115,116,117]. The initiation of innate immunity has an early onset and depends mostly on macrophages with the upregulation of adhesive molecules and proinflammatory cytokines, whereas adaptive immunity has a later onset and a dual effect on atherogenesis [118]. T lymphocytes facilitate the progression of atherosclerosis, but T helper 2 cells (Th2) and regulatory T cells (Treg) can dampen the process through the production of transforming growth factor (TGF)-β and interleukin (IL)-10 [114,119]. B lymphocytes have conflicting functions, since B1 cells perform an atheroprotective effect, while B2 cells are pro-atherogenic [116,119]. Zinc, an important trace element, is crucial to the function of immune cells. In the acute phase of inflammation, zinc undergoes a transient redistribution from serum to organs, especially the liver, which is mediated by the increased expression of ZIP-14 and up-regulation of MT in liver cells. This transient redistribution leads to temporary hypozincemia and upregulation of IL-1β, tumor necrosis factor (TNF)-α, S100 calcium-binding protein A8 (S100A8), and matrix metalloproteinase-8(MMP-8) [120]. Long before plasma zinc concentration changes, immune cells have the ability to respond to zinc deficiency early on [121]. The potential mechanism of how zinc influences the immune cells and the progress of atherosclerosis is explained below.

#### 3.2.1. Zinc, Macrophage, and Cytokines

OxLDL is characterized by the properties of damage-associated molecule patterns (DAMPs) and can interact with scavenger receptors or toll-like receptors, triggering an inflammatory response [116,117,122]. Circulating myeloid-derived monocytes migrate and relocate in the intima to differentiate into macrophages with a pro-inflammatory phenotype. Zinc affects the phagocytosis and inflammatory function of macrophages through a series of regulatory actions [123] and acts as an intracellular second messenger in signaling pathways [124]. Dubben et al. [125] found that zinc has a specific negative impact on monocyte differentiation, which is not observed in granulocytes. Zinc chelator TPEN can promote monocyte differentiation through augmentation of cyclic adenosine monophosphate (cAMP) production, which is formerly decreased by an inhibitive function of zinc ion on the cAMP-synthesizing enzyme adenylate cyclase. Another study indicates that zinc is the intermediary between monocyte adhesion to the vascular endothelium [126]. The excessive accumulation of cholesterol in macrophages activates the inflammasome NOD-like receptor thermal protein domain associated protein 3 (NLRP3), a protein that cleaves IL-1β precursor into its active form and enhances the expression of cytokines [116]. Zinc supplementation can reduce NLRP3 inflammasome activation [127], whereas TPEN inversely activates NLRP3 and improves the levels of pro-inflammatory cytokines, such as IL-1β and IL-6 in macrophages [128]. 

Inflammatory macrophages facilitate the progression of atherosclerosis and cause expansion of the lesion, leading to plaque morphological changes, eventually resulting in plaque rupture and acute lumenal thrombosis via the release of pro-inflammatory cytokines [122]. Zinc deficiency influences the generation of cytokines, and cytokines, in turn, affect zinc distribution. IL-6 and IL-1β are the major pro-inflammatory cytokines that innate immune cells secrete [124]. A meta-analysis conducted by Faghfouri et al. [129] indicated that zinc supplementation can decrease IL-6 levels, and another former study shows that IL-6 induces low zinc ion availability by modulating MT gene expression [130].

#### 3.2.2. Zinc and Lymphocytes

Along with macrophage and dendritic cell accumulation, adaptive immunity joins to exert a deleterious influence on plaque formation. Studies confirmed that the absence of T cells and B cells, which were induced by recombination-activating genes (RAGs) deficit, demonstrated alleviated atherosclerosis in *Apoe^–/–^ or Ldlr^–/–^* mice [131]. T cells play a predominant role in disease progression and constitute 10% of total plaque cells, 70% of which are CD4+ T cells, with B cells only occasionally found in plaque [119]. Atherosclerosis is known as a Th1 disease, as Th1 cells are the major producer of proatherogenic cytokines, such as interferon (IFN)-γ, IL-2, IL-3, and TNF. Zinc deficiency in the experimental human model results in imbalanced Th1/Th2 function. The products of Th1 cells, such as IL-2 and IFN-γ, decline under zinc deficit while Th2 cell products (such as IL-4, IL-6, and IL-10) remain unaffected [132,133]. 

Zinc contributes to T cell regulation in two ways: the signaling pathway initiated by T cell receptor (TCR) or by cytokine stimulation [124]. In the TCR activation-mediated signaling pathway, the CD3 subunits of the TCR complex are phosphorylated by lymphocyte-specific protein-tyrosine kinase (LCK) and promote the assembly of the immunological synapse, leading to the activation of the downstream signaling cascade [134]. Zinc facilitates the process by regulating the activity of LCK in multiple ways and plays a vital role in T cell development and activation [135,136]. After the TCR trigger, certain zinc transporters, such as ZIP-6 and ZIP-8, are up-regulated and induce a rise in cellular zinc concentration, which is directly involved in T cell activation [137,138]. On the other hand, the cytokine-mediated signaling pathway is regulated by IL-2, and according to Faghfouri et al., zinc can increase IL-2 levels [129]. IL-2 stimulation can induce T cell proliferation and development through three different signaling pathways: Janus kinase 1/3 (JAK1/3)- signal transducer and activator of transcription 5 (STAT5), ERK1/2, and phosphoinositide 3-kinase (PI3K)/protein kinase B (Akt) [124]. The interaction between zinc and immune cells in atherogenesis has not been directly confirmed through animal model studies. Further research should be conducted in this regard. 

## 4. Role of Zinc in Risk Factors of Atherosclerosis

In addition to the direct role of zinc in atherosclerosis, zinc furthermore exerts important positive or negative functions via various risk factors of the atherogenic process, such as lipid metabolism, glucose metabolism, and blood pressure. 

### 4.1. Role of Zinc in Lipid Metabolism

In humans, adipose tissue is the main lipid repository and the interaction between zinc and adipose tissue is critical in lipid metabolism. Olechnowicz et al. [90] previously proposed that zinc may regulate lipid metabolism by interfering with leptin production. However, other studies found no significant association between zinc supplementation and overall serum leptin [139,140]. A recent study proposed that zinc could induce lipolysis by activating the zinc/MTF-1/PPARα and calcium/calmodulin-dependent protein kinase kinase-β (CaMKKβ)/adenosine 5′-monophosphate-act protein kinase (AMPK) pathway, which further induce autophagy-mediated lipophagy [141]. Zinc-α2-glycoprotein (ZAG), an adipokine, plays a critical role in lipid metabolism, including reducing fatty acid synthesis, mobilizing lipids, and stimulating lipolysis [142]. The expression of ZAG was significantly reduced in the liver and adipose tissue of the obese population [143]. A zinc-binding site exists in the structure of ZAG [144], which suggests that the effect of zinc on lipid metabolism may be mediated by ZAG. In addition, zinc prevents the oxidation of LDL, stopping the main mechanism of atherogenesis [145].

High zinc levels increase hepatocyte activity and improve hepatic lipid metabolism in rats, while zinc deficiency impairs hepatic lipid metabolism [146]. Zinc deficiency in the prenatal and postnatal life of Wistar rats was shown to increase hypertriglyceridemia and adipocyte hypertrophy and adequate dietary zinc in postweaning growth could partially reverse this change [147]. Zinc supplementation could reverse abnormal lipid metabolism and elevated liver enzymes (such as aspartate aminotransferase and alanine transaminase) found in high-fat diet (HFD)-fed rabbits [148]. In lipid profiles, dietary zinc supplementation significantly reduced cholesterol and cholesterol oxidation products in rat serum [149]. In humans, obesity and being overweight were associated with decreased serum zinc levels [150]. Zinc deficiency or marginal zinc deficiency was associated with lower lipid profiles in healthy individuals [151]. However, the effects of zinc on specific lipid-conjugated molecules remain contradictory. A cross-sectional study on 7597 United States adults found that zinc intake was inversely associated with the ratio of total cholesterol (TC) to high-density lipoprotein (HDL) cholesterol [152]. However, Yary et al. [153] noticed an inverse relationship between serum zinc level and HDL cholesterol in middle-aged and older eastern Finnish men. In a meta-analysis, Khazdouz et al. [154] indicated that zinc supplementation could significantly decrease the plasma level of triglyceride, TC, and very-low-density lipoprotein (VLDL). Recently, Saykally et al. [155] noticed that the reduced amount of ZEB1 could increase adiposity during early weight gain in female TCF8+/- mice fed with HFD. Sterol regulatory element-binding protein 2 (SREBP-2), a master transcriptional regulator of cholesterol biosynthetic genes and LDL receptor genes, is highly involved in foam cell formation and atherogenesis [156]. Overexpression of ZEB1 could inhibit the endogenous expression of SREBP-2, thereby regulating the metabolism of cellular cholesterol [157]. Evidence also suggested that ZEB1 can regulate the expression of ATP-binding cassette transporter A1 (ABCA1) to reduce lipid accumulation in macrophages [55]. KLF2 is another important zinc finger transcription factor involved in macrophage form cell formation. Atkins et al. [52] noted that hemizygous deficiency of KLF2 increased diet-induced atherosclerosis in ApoE-deficient mice, indicating its atheroprotective role.

### 4.2. Role of Zinc in Glucose Metabolism

Zinc performs critical functions in insulin secretion and regulation of glucose metabolism. Zinc deficiency was associated with decreased insulin secretion from pancreatic β cells [158]. Filios et al. [159] indicated that knockdown of ZEB1 could significantly increase the apoptosis of pancreatic β cells. Zinc is believed to modulate the effects of insulin through different mechanisms [158,160]. Furthermore, serum zinc level was inversely associated with insulin resistance [161]. Several zinc homeostasis-associated proteins expressed in adipocytes, including ZnT-7, ZIP-13, and ZIP-14, can influence adipogenesis and the function of adipocytes, thereby participating in insulin resistance [158,162]. Apart from the indirect regulation of glucose metabolism via insulin regulation, zinc also directly regulates glucose metabolism. Studies have shown that zinc plays a vital role in the inhibition of gluconeogenesis, as well as the stimulation of glucose uptake and glycogen synthesis [163]. Moreover, it should be noted that a complex correlation exists between zinc, insulin, and diabetes. Diabetes can affect zinc homeostasis, by increasing the urinary loss of zinc and decreasing the total bodily zinc level [164].

Sisnande et al. [165] found that zinc restriction in male Swiss mice was associated with smaller pancreatic islets, mediated by increased apoptosis. Zinc supplementation could enhance the function of pancreatic β cells in HFD-fed mice, thereby improving glucose homeostasis [166]. Zinc supplementation in mice with HFD-induced liver injury could reduce glucose production and lipid deposition and promote glucose absorption [167]. These findings suggested that high zinc levels may alleviate the abnormal metabolism of glucose and lipid induced by HFD. In prediabetic populations, zinc supplementation significantly improved pancreatic β-cell function and decreased fasting blood glucose (FBG), 2 h postprandial glucose levels in oral glucose tolerance test, and insulin resistance [168]. A recent meta-analysis of randomized controlled trials (RCTs) focusing on the effects of zinc supplementation on cardiometabolic risk factors also indicated that zinc supplementation significantly decreases FBG and hemoglobin A1c (HbA1c) levels [154]. However, the association between serum zinc and type 2 diabetes mellitus (T2DM) remains controversial, with an inverse association [169], positive association [170,171], and no association [172] found in different studies. Fernández-Cao et al. [173] carried out a meta-analysis and found that slightly elevated dietary zinc intake may prevent the development of T2DM, while the elevation of serum zinc level might increase the risk of T2DM. Future studies focusing on the role of dietary or serum zinc levels in T2DM pathophysiology are warranted.

### 4.3. Role of Zinc in the Regulation of Blood Pressure

Zinc may regulate blood pressure (BP) through different mechanisms. Firstly, copper-zinc superoxide dismutase (Cu/Zn-SOD) conducts an important function in regulating hypertension by influencing oxidative stress. Studies have demonstrated that both excessive and insufficient zinc intake inhibits the activity of Cu/Zn-SOD [174,175]. Secondly, zinc may regulate BP by regulating the renin-angiotensin-aldosterone system (RAAS). Zinc is an active component of the angiotensin-converting enzyme (ACE), and its deficiency decreases serum ACE activity [176]. During critical development periods of Wistar rats, zinc deficiency was associated with morphological renal alterations and long-term/permanent changes in the renal RAAS system [177]. Thirdly, renal Na+-Cl- cotransporter (NCC) is a zinc-regulated transporter, and zinc deficiency could upregulate NCC expression and renal Na^+^ transport, contributing to an increase in BP [178]. In addition, NO is a key molecule involved in the tonic relaxation of systemic vasculature and BP regulation [179]. As previously described, zinc deficiency decreases the expression and activity of NOS in the vascular and renal system, leading to a decrease in NO activity [180]. Finally, zinc may also regulate BP by influencing intracellular calcium homeostasis [181].

Zinc may regulate BP in a bidirectional way. Some animal studies found that zinc restriction was associated with higher systolic BP (SBP) [147], while others indicated that excessive zinc intake also significantly elevated SBP, presumably through superoxide-induced oxidative stress [182]. The results from human studies focused on the relationship between zinc and hypertension are also inconsistent. A recent meta-analysis on 9 RCTs with 544 patients found that zinc supplementation significantly reduces the SBP but has no significant effect on diastolic BP (DBP) [183]. Several clinical trials and population-based studies found that zinc deficiency is associated with prehypertension in apparently healthy subjects [184], and zinc supplementation is beneficial for BP regulation [168,185]. Other studies found no association between zinc supplementation or serum zinc level and SBP [186,187,188,189]. All these findings suggested that both a lack and excess of zinc intake may have an adverse effect on elevated BP, indicating a U-type or J-type relationship between zinc and hypertension. Additional high-quality clinical trials focusing on the effects of zinc intake on blood pressure are warranted for further evidence.

## 5. Zinc and Atherosclerosis: Animal and Human Studies

### 5.1. Animal Studies

Animal studies found that zinc deficiency was associated with increased plasma concentration of triacyl glycerides and cholesterol and a higher lipoprotein-cholesterol distribution toward the non-HDL fraction [190,191]. ApoE knock-out (AEKO) mice with suboptimal dietary zinc intake were associated with raised proatherogenic lipoprotein levels and developed more aortic plaques than mice consuming adequate zinc [192]. In atherogenic mouse models, zinc supplementation suppressed the abnormal plasma lipid profile and the proinflammatory events in atherosclerosis [190]. Studies also found that dietary zinc supplementation significantly reduced total cholesterol accumulation in the aorta and inhibit the development of aortic atherogenesis in HFD-fed New Zealand rabbits [193,194,195,196]. A recent study conducted by Cheng et al. [197] suggests the role of zinc influx in monocyte adhesion and recruitment. Ex vivo and AEKO mice model experiments confirmed ZIP8 overexpression and increased zinc influx in monocyte adhesion to aortas and nascent atherosclerosis lesions, which potentially promote the inflammatory response towards arterial walls. Ren et al. [194] further found that zinc levels in aortic atheromatous lesions of zinc-supplied rabbits were not significantly different from that in control rabbits. However, zinc supplementation was associated with significantly lower iron lesion levels. Based on these findings, they proposed that the antiatherogenic effect of zinc may be mediated by inhibiting iron-catalyzed free radical reactions. Several studies indicate the anti-apoptosis function of zinc in VSMCs [109] and the detrimental effect of zinc deficiency on vascular health [198] (see Table 1). A variety of zinc finger proteins also exhibit significant influence in the pathogenesis of atherosclerosis by regulating the transcription process. Sayin et al. [199] found that zinc finger protein 148 (ZFP148) deficiency exerted an atheroprotective effect by increasing the activity of p53 and inhibiting the proliferation of macrophages in AEKO mice. KLF14, like other transcription factors in the KLF family, shows an anti-atherosclerotic effect in ApoE^−/−^ mice by promoting cholesterol efflux and inhibiting inflammatory response [200]. In addition, POZ/BTB and AT-hook-containing zinc finger protein 1 (PATZ1) and Juxtaposed with another zinc finger gene 1 (JAZF1) also play a protective role in atherosclerosis [201,202].

### 5.2. Human Studies

Epidemiological studies of zinc deficiency and atherosclerosis are scarce and most human studies are cohort studies and RCTs. Several studies focusing on zinc and vascular physiology indicated an inverse relationship between dietary zinc intake and carotid intima-media thickness [203,204] or aortic calcification [205]. A recent study conducted by Dziedzic et al. [206] measured the zinc level of hair samples from 133 CAD patients that were evaluated by angiography. A lower level of zinc was detected in the hair of patients with T2DM and high serum TG concentration. However, there are no significant statistical differences among groups with varying severity of CAD, which is in contrast with many other studies.

A pilot study found an association between the low ratio of serum zinc to 24 h urine zinc and severe coronary atherosclerosis in angiography [207]. The incidence of cardiovascular disease-related events was reported to be influenced by the level of both dietary zinc intake [208] and serum zinc [209]. A systematic review found that a lower incidence of CVDs was associated with higher serum zinc, but not higher dietary zinc intake [221]. The difference in the conclusions of these studies might be partially explained by the difference in sources of dietary zinc. Higher dietary zinc intake could elevate the level of serum zinc, however, a poor correlation exists between the level of serum zinc and zinc intake, since zinc intake from various sources varies greatly in its bioavailability and metabolism [222,223]. Coronary artery calcium progression is considered an important marker of subclinical coronary atherosclerosis. In clinical CVDs-free populations, Gao et al. [210] found an association between a lower risk of coronary artery calcium progression and higher dietary zinc intake from non-red meat sources. De Oliveira Otto et al. [211] reported a positive association between higher dietary zinc intake from red meat sources and the risk of CVDs. Considering the inconsistency between dietary zinc intake and serum zinc, it may be beneficial to investigate how zinc bioavailability is associated with atherosclerosis. Phytate/zinc molar ratio is inversely associated with zinc bioavailability. Jung et al. [212] found that the molar ratio of phytate to zinc was positively associated with the risk of atherosclerosis in women aged ≤ 65. This finding might suggest an inverse relationship between zinc bioavailability and the risk of subclinical atherosclerosis. 

Zinc deficiency has been recognized to function negatively in atherogenesis. Compared with healthy controls, patients with established atherosclerosis were associated with a significantly lower serum zinc level, which further decreases in patients with unstable angina [213,214]. Zinc supplementation was shown to be a promising treatment in patients with severely symptomatic and inoperable atherosclerotic disease [224]. Lee et al. [215] found that among alcohol drinkers who consumed ≥ 10 g of alcohol per day, CVD-associated mortality was inversely associated with dietary zinc intake. A prospective study on healthy Japanese men and women without CVDs found that higher dietary zinc intake was inversely associated with CVD-associated mortality in men but not women [216]. However, a ten-year follow-up study in Jiangsu province reported no significant relationship between relative zinc intake and CVD-associated mortality [217]. RCTs have been conducted to verify the positive effects of zinc. Bao et al. [218] found that increased plasma zinc level is associated with decreased plasma high-sensitivity C-reactive protein (hsCRP), IL-6, VCAM-1, secretory phospholipase A2 (sPLA), malondialdehyde, and hydroxyalkenals (MDA+HAE). Furthermore, zinc supplementation decreased the production of TNF-α, IL-1β, VCAM-1, and MDA+HAE, as well as the activation of NF-κB, while an increase in anti-inflammatory proteins A20 and PPAR-α were found in human monocytic leukemia THP-1 cells and human aortic endothelial cells. Costarelli et al. [219] confirmed these findings about inflammatory molecules via a zinc deficit clinical trial on overweight/obese adults. Results indicated that the zinc deficit group presented higher inflammatory markers, such as α 2-macroglobulin (A2M) and CRP, and lipid assets, such as TC, LDL, and triglycerides, in parallel with decreased zinc homeostasis genes (SLC30A-1, MT-1A, and MTF-1) and up-regulated inflammatory genes (IL-6, IL-1α, IL-1β, and A2M). However, a later study by Seet et al. [220] indicated that oral zinc supplementation did not exert change in markers of oxidative damage and vascular function. More RCTs are required for further understanding and instruction of zinc supplementation and patient benefits (see Table 1).

## 6. Limitations and Future Perspectives

Zinc is an indispensable micronutrient involved in multiple biological processes in organisms. Due to the lack of functional reserve for zinc, human zinc homeostasis primarily relies on daily dietary intake. The recommended dose of zinc intake is 11 mg/d and 8 mg/d for men and women, respectively, and should not exceed 40 mg/d [225]. Zinc deficiency is more commonly found in the elderly [226] and vegetarians [227] and usually progresses in a chronic and latent way, of which the manifestation is hard to connect with zinc homeostasis directly, much different from that of iron and iodine. The absence of a reliable and specific biomarker for the indication of in vivo zinc levels also restricts further study of zinc in clinical trials. The most adopted measurements for the estimation of adult zinc status at present are dietary zinc intake and plasma/serum zinc concentration [228]. Other indicators include hair, urinary, and blood cell zinc. However, none of them are able to represent an overall and long-term status of body zinc and zinc-binding proteins. Since zinc has an extensive and profound influence on cardiovascular physiology and cell metabolism, it is necessary to seek novel zinc status biomarkers that are less influenced by individual variability or develop an innovative technology to measure cellular zinc levels in clinical trials. 

All in all, the animal studies and epidemical research have indicated the potential negative role of zinc deficiency in atherosclerosis. However, specific pathways or interaction patterns between zinc and plaque formation or atherosclerotic progression remain unclear. In this review, we aim to investigate the underlying mechanism of how zinc affects atherogenesis progress through a review of the role that zinc plays in proatherogenic cells, such as ECs, VSMCs, and immune cells, as well as the risk factors, such as lipid metabolism, glucose metabolism, and blood pressure. Most evidence prompts a positive role of zinc in multiple physiologic processes, such as NO generation, anti-inflammation, anti-oxidation, and anti-apoptosis, while paradoxical effects are also indicated in immune cells. More research is required in this field, especially those that directly reveal the effect of zinc on atherosclerosis models, such as in vitro studies with atherosclerosis-specific stimuli, assays on plaque composition and morphological changes in animal models, or evaluation of plaque and stenosis degree in atherosclerosis patients. Therefore, further studies are needed to better elucidate the role of zinc in atherosclerosis. Taken together, zinc participates in the regulation of multiple cellular processes and may exert a pivotal role in atherogenesis and disease progression. Zinc supplementation is a potential adjunctive therapy for atherosclerosis and its complications. Further efforts should be expended on a comprehensive understanding of zinc and atherosclerosis and optimal prophylactic administration for the targeted populations.

## Figures and Tables

**Figure 1 biomolecules-12-01358-f001:**
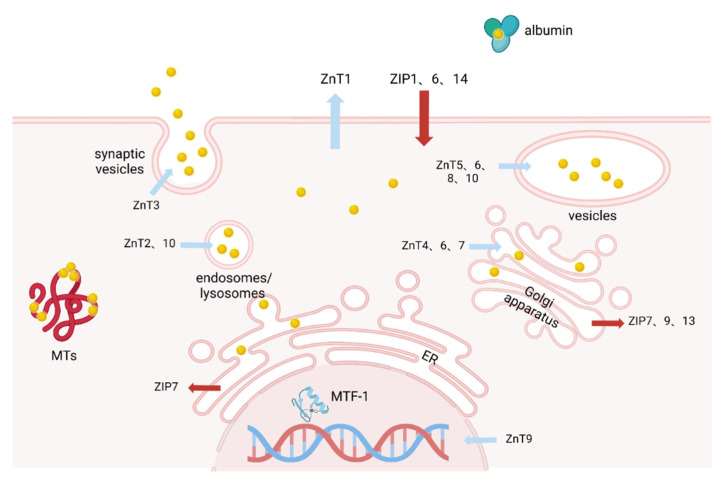
The localization of zinc transporters Zrt-, Irt-like proteins (ZIP, red arrow), Zn transporters (ZnT, blue arrow), metallothioneins (MTs), and metal response element-binding transcription factor-1 (MTF-1) in the cell.

**Table 1 biomolecules-12-01358-t001:** Animal and human studies that report the effects of zin on atherosclerosis.

Author, Year	Subjects	N	Study Design	Intervention	Outcome
** *Animal studies* **					
Reiterer et al. 2005 [190]	LDL receptor knock-out (LDL-R-/-) mice, 5 wk	11 (ZD) 11 (Z) 11 (ZS)	/	ZD: AIN-93 diets with 0 µmol Zn/g diet for 4 wk; Z: AIN-93 diets with 0.45 µmol Zn/g diet for 4 wk; ZS: AIN-93 diets with 1.529 µmol Zn/g diet for 4 wk	**Zinc deficiency was associated with:** - Higher mRNA expression of the antioxidant enzyme glutathione reductase, VCAM-1, and PPARγ; - Higher DNA binding activity of NF-κB in liver tissues; - Lower DNA binding activity of PPARγ. **Zinc supplementation was associated with:** - Reduced TG, total plasma cholesterol, and VLDL and HDL fractions of free cholesterol.
Shen et al. 2007 [191]	LDL receptor knock-out (LDL-R-/-) male mice, 5 wk	2 (ZD) 2 (Z)	/	ZD: low-fat (LF) diet with 0.4 mg/kg of zinc for 21 d, and then high-fat (HF) diet with 0.4 mg/kg of zinc for 1 wk; Z: low-fat (LF) diet with 33.1 mg/kg of zinc for 21 d, and then high-fat (HF) diet with 33.1 mg/kg of zinc for 1 wk	**Zinc deficiency was associated with:** - Higher total plasma cholesterol, total plasma fatty acids, and non-HDL fraction (VLDL, IDL, LDL); - Inhibited PPARγ transactivation activity induced by rosiglitazone.
Beattie et al. 2012 [192]	ApoE knock-out (AEKO) mice, 5 wk	24 (ZD) 12 (Z)	/	ZD: high saturated fat (21% w/w) and high cholesterol (0.15%) semi-synthetic diets containing suboptimal zinc (3 or 8 mg/kg) for 25 wk; Z: high saturated fat (21% *w*/*w*) and high cholesterol (0.15%) semi-synthetic diets containing adequate zinc (35 mg/kg) for 25 wk	**Zinc deficiency was associated with:** - Higher plaque area; - Elevated cytokine IL-1β, IL-6 and sVCAM-1; - Elevated total protein and cholesterol level; - Increased intramural calcium and phosphorous deposits (atheromatous plaque).
Alissa et al. 2004 [193]	Male New Zealand White rabbits, 6–10 wk	8 (ZS) 8 (C)	/	ZS: diet containing 0.25–1% (*w*/*w*) cholesterol plus zinc as its sulfate at 0.5% (*w*/*w*) for 12 wk; C: diet containing 0.25–1% (*w*/*w*) cholesterol for 12 wk	**Zinc supplementation was associated with:** - No significant difference in weight gain, integrated serum cholesterol levels, erythrocyte superoxide dismutase, hepatic enzyme activity; - Reduction in aortic atherogenesis lesion.
Ren et al. 2006 [194]	Male New Zealand white rabbits,/	6 (ZS) 6 (C)	/	ZS: zinc-supplemented diet SF03-017 (modified guinea pig and rabbit + 1% cholesterol + 1000 ppm (1 g/kg) zinc as zinc carbonate) for 8 wk; C: high-cholesterol diet (HCD) SF00- 221 (modified guinea pig and rabbit + 1% cholesterol) for 8 wk	**Zinc supplementation was associated with:** - No significant difference in total cholesterol, triglyceride, and LDL; - Lower HDL; - Decreased lesion areas from sections of aorta; - Decreased Fe lesion levels.
Abdelhalim et al. 2013 [195]	Male New Zealand white rabbits, 12 wk	5 (ZS) 5 (C)	/	ZS: zinc-supplemented high-cholesterol diet: NOR Purina Certified Rabbit Chow no. 5321 with 1.0% cholesterol plus 1.0% olive oil (100 g/day), with 350 ppm zinc for 12 wk; C: high-cholesterol diet: NOR Purina Certified Rabbit Chow no. 5321 with 1.0% cholesterol plus 1.0% olive oil (100 g/day) for 12 wk.	**Zinc supplementation was associated with:** - Delayed or retarded progression of atherosclerosis.
Jenner et al. 2007 [196]	Male New Zealand white rabbits,/	5 (ZS) 6 (C)	/	ZS: zinc-supplemented high cholesterol diet [GPR + 1% cholesterol + 1000 ppm (1 g/kg of diet) zinc as zinc carbonate] for 8 wk; C: High cholesterol diet [GPR + 1% cholesterol] for 8 wk.	**Zinc supplementation was associated with:** - No significant effect on levels of LDL or plasma triglyceride; - Lower total arterial wall cholesterol levels; - Decreased average lesion cross-sectional areas; - Decreased total F2-isoprostanes in plasma and arterial wall atherosclerosis; - Decreased cholesterol oxidation products in complete arterial wall.
Beattie et al. 2008 [198]	Sprague–Dawley rats, 100 g, 4 wk	Study 1: 9(MZD) 9(MZA) Study 2: 10(AZD) 10(APF) 10(AZA)	/	AIN-76 diet that contained 35, 6 or, 1 mg Zn/kg; MZD: 6 mg zinc/kg for 43 days; MZA: 35 mg zinc/kg for 43 days; AZD: 1 mg zinc/kg for 39 days; APF: 35 mg zinc/kg for 39 days, pair-fed with AZD group; AZA: 35 mg zinc/kg for 39 days.	**Zinc deficiency was associated with:** - Aorta cell structure, fatty acid metabolism, and carbohydrate metabolism; - Changes in smooth muscle contractility, insulin resistance, and perturbation of key Zn-dependent transcription factors; - Detriment to the maintenance of vascular health.
** *Human studies* **					
Yang et al. 2010 [203]	Middle-aged and elderly Korean populations, 40–89 y	4564	Cross-sectional analyses	/	- Zinc intake was inversely related to subclinical atherosclerosis.
Masley et al. 2015 [204]	Population attending an executive evaluation program, 23–65 y	592	Prospective cross-sectional analysis	/	- Lower intake of zinc was associated with higher mean carotid intima media thickness.
Chen et al. 2020 [205]	Noninstitutionalized civilian US populations, ≥40 y	1764 (AAC) 771 (noAAC)	Cross-sectional analyses	/	- Patients with severe AAC had a lower intake of dietary zinc; - A higher intake of dietary zinc was associated with lower odds of having severe AAC; - A higher serum zinc level was associated with higher supplemental zinc intake but not with dietary or total zinc intake; - Those with severe AAC had a higher intake of supplemental and total zinc.
Dziedzic et al. 2022 [206]	Patients with a history of previous MI who were treated with coronary Angioplasty, 37–95 y	133	Cross-sectional analyses	/	- No significant statistical differences in zinc levels between groups with different severity of CAD; - A negative correlation was identified between zinc content in the hair and serum TG concentration.
Giannoglou et al. 2010 [207]	Underwent diagnostic coronary angiography for evaluating chest pain. CAD: 66 ± 43 y; control: 61 ± 51 y	40 (CAD) 32 (control)	A pilot study	/	- No significant difference in serum zinc level; - CAD was significantly associated with higher urinary zinc and lower serum zinc/24-h urine zinc ratio; - Angiographic indices were positively associated with higher urinary zinc and negatively associated with serum zinc/24-h urine zinc ratio.
Milton et al. 2018 [208]	Mid-age Australian women, 50–61 y	9264	Large longitudinal study	/	- Dietary zinc intake was associated with a higher CVD incidence in women aged ≥ 50 years.
Mattern et al. 2021 [209]	Black and White participants, ≥45 y	2944	Population-based prospective longitudinal study	/	- Serum zinc levels were inversely associated with the incidence of ischemic stroke after adjustment for potential confounders.
Gao et al. 2021 [210]	Multi-ethnic participants, 45–84 y	6814	Prospective population-based observational cohort study	/	- Higher dietary zinc intake was significantly associated with a lower risk of CAC progression for women and men; - This association was found in higher dietary zinc intake from other sources, but not from red meat.
de Oliveira Otto et al. 2012 [211]	Adults free of clinical CVD from six U.S. communities, 45–84 y	6814	Prospective population-based study	/	- Zinc derived from red meat sources was associated with greater risk of incident metabolic syndrome; - Zinc from red meat, but not zinc from other sources, was associated with the risk of CVD.
Jung et al. 2013 [212]	Populations from multi-rural Korean communities, ≥40 y	5532	Cross-sectional analysis	/	- Phytate: zinc molar ratio was positively related to subclinical atherosclerosis risk in men, but not in women.
Alissa et al. 2006 [213]	Saudi male subjects with established CVD, 55.6 ± 12.1 y age-matched controls, 55.0 ± 11.6 y	130 (CVD) 130 (control)	Population-based study	/	- Urinary zinc was significantly lower in patients with CVD; - Serum zinc was negatively associated with atherosclerosis.
Qazmooz et al. 2021 [214]	Patients with atherosclerosis and age, sex-matched healthy controls	120 (atherosclerosis) 58 (control)	Machine learning studies	/	- Serum zinc decreased from controls to patients with atherosclerosis, then to atherosclerotic patients with unstable angina.
Lee et al. 2005 [215]	Postmenopausal women, 55–69 y	34,492	Population-based study	/	- In the total sample, dietary zinc intake was not associated with the risk of CVD mortality; - Among alcohol drinkers who consumed ≥10 g alcohol per day, dietary zinc showed an inverse association with CVD mortality.
Eshak et al. 2018 [216]	Middle-aged residents in 45 Japanese communities, ≥40 y	58,646	Population-based prospective cohort study	/	- Populations with higher zinc intake were less likely to be hypertensives but more likely to be diabetics; - Dietary zinc intake was inversely associated with mortality from CHD in men but not women.
Shi et al. 2018 [217]	Based on a subsample of the Chinese national nutrition and health survey representing Jiangsu province, ≥20 y	2832	Population-based prospective cohort study	/	- Zinc intake is positively related to all-cause mortality and there was no significant association between relative zinc intake and CVD mortality.
Bao et al. 2010 [218]	40 healthy elderly, 56–83 y	20(ZS) 20(C)	RCT	ZS: 45 mg zinc/d as gluconate for 6 mo; C: placebo for 6 mo	- Zinc supplementation increased plasma zinc concentration and decreased the concentrations of plasma hsCRP, IL-6, MCP-1, VCAM-1, sPLA, and MDA+HAE; - Zinc supplementation decreased the generation of TNF-α, IL-1β, VCAM-1, MDA+HAE, activation of NF-κB, and increased anti-inflammatory proteins A20 and PPAR-α in human monocytic leukemia THP-1 cells and human aortic endothelial cells.
Costarelli et al. 2010 [219]	Overweight/obese individuals (BMI ≥25 kg/m^2^), 43 ± 5 y	100 (Group 1) 123 (Group 2)	Clinical trial	Group 1: low-zinc dietary intake (<7 mg/day for females and <9.5 for males); Group 2: normal-zinc dietary intake (≥7 mg/day for females and ≥9.5 for males)	- Zinc deficiency group presents higher inflammatory markers (A2M and CRP) and - lipid assets (total and LDL cholesterol, triglycerides); - Zinc deficiency group shows decreased zinc homeostasis genes (SLC30A1, MT-1A, and MTF-1) and up-regulated inflammatory genes (IL6, IL1-α, IL1-β, and A2M).
Seet et al. 2011 [220]	Male T2DM patients, ≥21 y	20(ZS) 20(C)	RCT	ZS: 240 mg zinc/day as gluconate for 3 mo; C: 2 tablets of placebo (99% microcrystalline cellulose, 1% magnesium stearate) for 3 mo	- Oral zinc supplementation did not exert change in markers of oxidative damage and vascular function.

**Abbreviations:** A2M: alpha 2-macroglobulin; AAC: abdominal aortic calcification; APF: pair-fed control group for acute zinc deficiency; AZA: zinc adequate control group for acute zinc deficiency; AZD: acute zinc deficiency; C: without zinc supplementation; CAC: coronary artery calcium; CAD: coronary artery disease; CHD: coronary heart disease; CVD: cardiovascular diseases; HDL: high-density lipoprotein; hsCRP: high-sensitivity C-reactive protein; IDL: intermediate density lipoprotein; IL: interleukin; MCP-1: macrophage chemo-attractant protein 1; MDA+HAE: malondialdehyde and hydroxyalkenals; MT: metallothionein; MTF-1: metal response element-binding transcription factor-1; MZA: zinc adequate control group for marginal zinc deficiency; MZD: marginal zinc deficiency; NF-κB: nuclear transcription factor kappa B; PF: pair-fed control group; PPAR: peroxisome proliferator-activated receptor; TG: triacylglycerides; TNF-α: tumor necrosis factor α; SLC30A: solute carrier 30A; sPLA: secretory phospholipase A2; VCAM-1: vascular cell adhesion molecule-1; VLDL: very low-density lipoprotein; Z: adequate zinc; ZD: zinc deficiency; ZS: zinc supplementation.

## Data Availability

Not applicable.

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
