# Peer review of "Investigating the Role of Zinc in Atherosclerosis: A Review"

_biomolecules, 2022, doi:10.3390/biom12101358_

Round 1

Reviewer 1 Report

The authors chose the title “Role of zinc in atherosclerosis”. They described many references to show that, but the direct interaction between atherogenesis and zinc status has still not been discussed. This review does not elucidate the relationship between zinc and atherosclerosis nor the mechanism of how zinc homeostasis-related proteins interplay with atherogenic progress. The evidence does not reveal the primary role of zinc deficiency in association with atherogenesis. It might present a discussion about zinc’s essential role in cell physiology. It would need much work to change the title and the history.

The review mentions a few references about the direct correlation of zinc with atherosclerosis. There are many references quoting the essential role of zinc in cellular homeostasis in cells relevant to atherosclerosis, but a few in atherosclerosis animal models and in vitro using oxidized LDL, for instance. There is a discussion about the relation between zinc and atherosclerotic risk factors. Most of the references and discussions are indirect and may be circumstantial. The authors quoted: “The interaction between zinc and immune cells in atherogenesis has not been directly confirmed through animal model studies”.

In many parts, the authors mention unclear mechanisms as: “are major contributor to atherosclerosis progression” (line 160), “critically influences the development of atherosclerosis” (line 164), “anti-atherosclerotic effects” (line 168-169), “NO is shown to be involved” (line 185), “NF-kB plays an important role in inflammation, and the development of atherosclerosis (line202), “apoptotic cell death is commonly involved in atherosclerotic plaque” (line 226). How, when, and specifics are missing. Detail description is a must. Why is the role of zinc in LPS-induced apoptosis discussed? Why is it relevant to atherosclerosis?

Table 1 shows some of the effects of zinc in atherosclerosis, but for instance, the first and second references are not about atherosclerosis.

The detailed physiopathology of atherosclerosis is missing. The reader must understand the whole process first and then read about the specific role of zinc in each part of the process, each cell type, and risk factors.

There are new references that must be added. The titles of the topics are general (3.2.1, 3.2.2, 4.1)

A current and critical view of the main subject is also missing.

Reviewer 2 Report

The authors aimed to elucidate the relationship between zinc and atherosclerosis, as well as the mechanism of how zinc homeostasis-related proteins interplay with atherogenic progress, thereby revealing the primary role of zinc deficiency in association with atherogenesis.

The paper is poorly written. There is no indication of the database searched, the keywords used, the methodology of searching relevant articles in the current literature.

While the authors comment some findings, I think the article is quite incomplete. Moreover, this article sounds like a book chapter rather than a review article.

Overall considered, I do not think it is suitable for publication in this journal.

Reviewer 3 Report

In this narrative review, the authors summarize the current evidence on the role of zinc in preventing or attenuating atherosclerosis. The subject is of great interest, and the manuscript is clear and well written. Minor comments are reported below.

Line 124. The authors could use the acronym “CVDs” for “cardiovascular diseases”.

Lines 223-236. There is an excessive use of the verb “found”. In general, in the text, the verb “to find” and “to indicate” were excessively used.

Line 509. Please change to “the risk of..”

Round 2

Reviewer 2 Report

amended manuscript is acceptable